# Nash Up, Virus Down: How the Waiting List Is Changing for Liver Transplantation: A Single Center Experience from Italy

**DOI:** 10.3390/medicina58020290

**Published:** 2022-02-14

**Authors:** Alberto Ferrarese, Sara Battistella, Giacomo Germani, Francesco Paolo Russo, Marco Senzolo, Martina Gambato, Alessandro Vitale, Umberto Cillo, Patrizia Burra

**Affiliations:** 1Department of Gastroenterology, Verona University Hospital, 37124 Verona, Italy; alberto.ferrarese17@gmail.com; 2Multivisceral Transplant Unit, Department of Gastroenterology, Padua University Hospital, 35100 Padua, Italy; sarabattistella93@gmail.com (S.B.); germani.giacomo@gmail.com (G.G.); francescopaolo.russo@unipd.it (F.P.R.); marcosenzolo@hotmail.com (M.S.); martina.gambato@gmail.com (M.G.); 3Liver Transplant Unit, Department of Hepatobiliary Surgery, Padua University Hospital, 35100 Padua, Italy; alessandro.vitale@unipd.it (A.V.); cillo@unipd.it (U.C.)

**Keywords:** cirrhosis, NAFLD, hepatocellular carcinoma

## Abstract

*Background and Objectives*: Non-alcoholic steatohepatitis (NASH) has become the leading indication for liver transplantation in many countries, with a growing rate in the Western world. NASH patients are older and share a higher risk of comorbidities and cancer than patients with viral and/or alcoholic etiologies. The aims of this study were to evaluate waiting list (WL) registration and liver transplantation rates in patients with NASH-related cirrhosis at Padua University Hospital in the last fifteen years (1.2006–6.2020) and to compare clinical characteristics and indications for liver transplantation between patients with and without NASH, as well as the WL survival and post-transplant outcome. *Materials and Methods*: All adult patients with cirrhosis listed for liver transplantation at Padua University Hospital between 1.2006 and 6.2020 were retrospectively collected using a prospectively updated database; patients with NASH-related cirrhosis were divided by indication for liver transplantation (Dec-NASH vs. hepatocellular carcinoma (HCC)-NASH) and compared with patients with other etiologies of liver disease. The outcomes in terms of waiting list survival and post-transplant outcome were assessed. *Results*: One thousand four hundred and ninety-one adult cirrhotic patients were waitlisted during the study period. NASH patients accounted for 12% of all WL registrations, showing an increasing trend over time (from 2.5% in 2006 to 23% in 2020). In the last five years, NASH was the third, but most rapidly growing, indication for liver transplantation at our center. This trend was confirmed both for patients with decompensated cirrhosis (from 1.8% to 18%) and HCC as leading indication for transplantation (from 4% to 30%). NASH patients were older than non-NASH ones (mean ± SD age 59 ± 9 vs. 56 ± 9 years; *p* < 0.01), whereas no difference was found in gender or Child-Pugh of the model for end-stage liver disease score at WL registration. A majority (60.9%) of NASH patients underwent liver transplantation, showing 1-, 5- and 10-y post-transplant survivals of 86%, 73% and 60%, respectively. *Conclusion*: NASH cirrhosis has become a rapidly growing indication for liver transplantation at our center, both for HCC and decompensated disease, with good post-transplant survival.

## 1. Introduction

The landscape of liver transplantation has significantly changed over the last decades, in view of significant therapeutic advances in Hepatology. As a matter of fact, there has been a huge decrease in waitlistings for patients with hepatitis C (HCV)-related cirrhosis after the advent of direct-acting antivirals [1,2,3]. On the other hand, Western countries are having a growing increase of cirrhotic patients with non-alcoholic steatohepatitis (NASH). Therefore, the need for liver transplantation for such patients, both for decompensated disease and hepatocellular carcinoma (HCC) as the primary indication, has shown a persistently increasing trend [4,5,6].

The prevalence of NASH is heterogeneously reported worldwide [7]. It seems to be less pronounced in Southern Europe, where a different prevalence in risk factor profiles and the presence of the Mediterranean diet have been considered plausible protective factors in the past. Nevertheless, few data have been available up to now on the rising trend of NASH in the liver transplant setting at our latitude.

Patients with NASH cirrhosis are often older at the time of waitlisting than patients with other etiologies and share more comorbidities (e.g., cardiovascular diseases and chronic kidney disease) [8,9]. Liver transplant candidates with HCC arising on NASH often have larger tumors, probably because they are not included in surveillance programs [10,11,12]. Nevertheless, according to recent data, their post-transplant survival is comparable with that of patients sharing other etiologies [13].

These evolutions involve significant implications for the management of the liver transplant candidate, as well as for the post-operative follow-up. The aim of this study was therefore to evaluate the change in waitlist registrations for NASH cirrhosis at a single center institution in Italy, describing the temporal trends over a long time period, to evaluate the outcome of such patients while on the waiting list and to compare them with that of patients with other disease etiologies and to describe their post-transplant outcomes.

## 2. Materials and Methods

This was an observational, single-center study. All patients registered on the waiting list for liver transplantation at Padua University Hospital between January 2006 and June 2020 were retrospectively included from a prospectively updated database. Patients with acute liver failure, with indications for transplant other than cirrhosis, retransplants or patients younger than 18 years of age were excluded from the study. None of the patients underwent AB0-incompatible liver transplantation or living donor liver transplantation. A diagnosis of NASH cirrhosis was made according to the current guidelines [14,15], whereas patients with mixed etiology (e.g., alcoholic/viral and metabolic cirrhosis) were not included. For each transplant candidate, demographic characteristics (e.g., age, gender, Child-Pugh and model for end-stage liver disease (MELD) score) were collected during waiting list (WL) inclusion. Patients were further divided according to indication for WL registration: patients with HCC and compensated liver disease (e.g., MELD < 15; HCC group) and patients with decompensated disease independently of the presence of HCC (e.g., MELD ≥ 15 or complications of portal hypertension; dec group). Patients with a diagnosis of HCC received waiting list prioritization according to a previously published policy recently updated [16]. A tacrolimus-based immunosuppressive regimen was usually adopted in the post-LT period, adding mycophenolic acid or everolimus as CNI-sparing agents. Steroids were not used as long-term immunosuppressive agents. The immunosuppression schedule was tailored according to pre-LT and post-LT comorbidities and complications (e.g., chronic kidney disease, post-LT sepsis and tacrolimus-induced side effects). Outcome data were collected at the latest available follow-up. Given the retrospective design of this study, informed consent was waived. 

### Statistical Analysis

The categorical variables were calculated as frequencies and were compared using Fisher’s test. Continuous variables were calculated as means ± standard deviation (SD) or as medians (interquartile range, IQR), according to the variable distribution and were compared using the Student’s *t*-test or Mann–Whitney *U* test, as appropriate. *p*-values < 0.05 were considered statistically significant. Survival analyses were performed using Kaplan–Meier curves (log-rank test). Analyses were performed using SPSS software version 22 (Chicago, IL, USA).

## 3. Results

One thousand four hundred and ninety-one adult cirrhotic patients fulfilling the inclusion criteria were retrospectively retrieved from our prospectively updated database. The clinical characteristics at the time of WL registration are depicted in Table 1. In detail, the patients were mostly men (77%) with a mean ± SD age of 56 ± 9 years. Overall, the WL registrations rose from 80 patients/year in the period 2006–2011 to roughly 100 patients/year in the period 2012–2019. Decompensated cirrhosis (dec-group) remained the most frequent indication for WL registration, even though there was a stable increase of HCC as the primary indication (Appendix A).

One hundred and seventy-nine patients had NASH-related cirrhosis as the underlying disease, accounting for 12% of the overall population. There was an increasing trend in WL registrations for NASH in the last five years. While, in 2006, NASH accounted for only 2.5% of all WL registrations, it became the third indication in 2020, being 23% of all registrations, with a prevalence similar to HCV and alcohol-related disease (25% and 27%, respectively). Notably, since 2017, among waitlisted patients, more than 20% per year had NASH cirrhosis, showing a stable trend over time. Moreover, from 2006 to 2020, NASH demonstrated the greatest increase as a cause of chronic liver disease among new liver transplant WL registrations (Figure 1). 

When patients with NASH cirrhosis were compared with the others (Table 1); they were older at the time of WL registration (mean age 59 ± 9 vs. 56 ± 9; *p* < 0.01) and showed a higher body mass index (27 ± 5 vs. 25 ± 4; *p* < 0.01), whereas no difference in gender, portal vein thrombosis, refractory ascites or severity of liver disease in terms of the Child-Pugh score and MELD score was found. Notably, NASH patients had a nonsignificant higher prevalence of HCC than the control group (53.6% vs. 50.1%). 

### 3.1. Stratification according to Indication for WL Registration

Figure 2a describes the trends in WL registration for patients with a decompensated disease (dec group). Overall, there was a decrease of WL registrations for HCV, especially after 2015. The rate of patients with alcohol-related diseases accounted for 30% of all registrations, with a steady course, becoming the first indication for transplantation in such a cohort in the last 5 years. NASH steadily grew from 1.8% in 2006 to 18% in 2020, becoming the third indication for transplantation—behind alcohol and hepatitis B but ahead of hepatitis C—in the last three years. In fact, the increase in registrations for NASH had already been noted before the advent of direct-acting antivirals, since it grew by 200% from 2006 to 2012. Furthermore, it reached a greater spike in the last 5 years, with a further 80% increase from 2012 to 2020.

We then explored the characteristics of patients whose indication for liver transplantation was HCC. As expected, the rate of waitlisted patients for this indication with a viral cirrhosis decreased from 75% in 2006 to 55% in 2020, mainly after the fall of hepatitis C. There was a significant increase in WL registration for patients with HCC arising from NASH (from 4% in 2006 to 30% in 2020). Indeed, in 2020, NASH became the leading indication for liver transplantation for HCC in compensated disease at our center, with a rate similar to that of hepatitis B but ahead of hepatitis C and alcohol (25% and 15%, respectively; Figure 2b). 

### 3.2. Period on the Waiting List for Patients with NASH and Post-Transplant Survival

We then looked at the outcome of patients with NASH during their period on the waiting list. The mean waiting time for the entire cohort was 11 ± 15 months; at the time of analysis (31 October 2021), three (1.6%) patients were still on the WL; 5 (2.8%) patients were excluded due to the improvement of their liver disease (1 for complete downstaging without HCC recurrence, 2 for improvement of ascites after trans-jugular intrahepatic portosystemic shunt placement and 2 for improvement of the biochemical profile and MELD persistently below 15) and 62 (34.6%) were removed or died due to disease progression. In detail, 19 (10.6%) HCC patients dropped out due to tumor progression outside the established criteria for liver transplantation; 32 patients (17.9%) died of liver-related conditions (including sepsis, variceal bleeding and hepatorenal syndrome) and 11 (6.1%) deaths were liver-unrelated. One hundred and nine patients (60.9%) underwent liver transplantation after a mean follow-up time of 8 ± 9 months, displaying a mean MELD score of 17.5 ± 7.6 at time of surgery. During the follow-up time, five (4.5%) patients underwent early re-transplantation for primary nonfunction, whereas none required late re-transplantation. The causes of death were sepsis (seven), primary non-function/early allograft dysfunction (five), liver-unrelated conditions (two), cardiovascular accidents (four) and HCC recurrence (four). 

On 31.10.21, after a median (IQR) follow-up time of 2.9 (1.1–4.72) years, the 1-, 3-, 5- and 10-year post-LT survivals were 86%, 77%, 73% and 60%, respectively (Figure 3). No difference in post-LT survival was found when patients were divided according to the primary indication for transplantation (*p* = 0.68 log-rank test; Appendix A). 

## 4. Discussion

In recent decades, two main events have dramatically changed the landscape of the liver transplant setting. First, the achievement of a sustained virological response after direct-acting antiviral therapy has been associated with an improvement of liver function and a reduction of HCC occurrence in patients with HCV-related advanced disease. Second was the spreading of NASH-related liver disease, especially in Western countries. In this paper, we explored the dynamics of these changes in the liver transplant setting at our center. 

We demonstrated a significant increase in waitlistings for NASH cirrhosis over time, from 2.5% in 2006 to 23% in 2020. This trend was consistent with recent data coming from European and non-European centers [5,17,18,19]. Moreover, we demonstrated that NASH became the most rapidly growing indication for liver transplantation in the last 5 years at our center, reaching annual rates historically applied only to alcohol and viral hepatitis. Finally, we demonstrated that this trend was equally present for patients both with HCC and decompensated cirrhosis as the primary indication for transplantation. Therefore, the first message of this paper is that the entire liver transplant program must be prepared for an epidemic of NASH, which 5 years ago seemed to be relevant only in transplant settings far from Europe, as in the US. 

It may be argued that the impressive fall in hepatitis C as an indication for liver transplantation significantly helped the rise of NASH. Nevertheless, hepatitis C still accounts for a relevant percentage of indications for transplantation, especially among patients with HCC. This has recently also been demonstrated in other studies [20]. Moreover, as depicted in Figure 2, the climb of NASH also became evident before the advent of direct-acting antivirals, reflecting the rise of this condition in the general population [7]. 

The NASH patients were older than the patients with other etiologies, displaying also a higher body mass index in our cohort. These two findings are consistent with the data already reported in the literature [21]. Even if not significant, NASH patients had a higher prevalence of HCC at the time of waitlisting. No difference was found between NASH and non-NASH candidates on the prevalence of refractory ascites or portal vein thrombosis. Notably, our data on portal vein thrombosis prevalence were similar to a recently published series from the US, including more than 3689 NASH transplant recipients [22]. A more hypercoagulable state in cirrhotic patients with NASH has been hypothesized, due also to the high presence of chronic inflammation and diabetes, with some evidence suggesting a higher prevalence than other etiologies. It is important to note, however, that we collected only thromboses at the time of WL registration; therefore, we are not able to assess the incidence of new cases occurring thereafter [23].

Sixty percent of patients with NASH underwent liver transplantation. Notably, more than one-third were excluded from the waiting list or died while awaiting liver transplantation. It is plausible that the older age and the presence of many cardiovascular and renal comorbidities may play a role. A comparison of the waitlist outcomes between NASH and non-NASH cohorts, after adjusting for comorbidities, will offer important information for the proper management of such patients. Finally, we explored the post-transplant survival of patients with NASH. We demonstrated a good 1-, 5- and 10-years post-transplant survival of 86%, 73% and 60%, respectively. These data are consistent with those recently published in a systematic review in this field [24]. A relevant number of patients died of sepsis (7) and of cardiovascular accidents (4); these two conditions accounted for half of all deaths in the post-transplant follow-up. Indeed, NASH confers a twofold higher risk of post-transplant death for sepsis and cardiovascular accidents, according to the recently published data [24].

The main strength of this study was the retrospective analysis of a prospectively collected database, which was routinely used to manage the waiting list over the whole time period. We were therefore able to provide granular data about each of our waitlisted patients. Second, the data came from a large Transplant Center in Northern Italy, with a well-recognized transplant program. However, this study had some limitations that need to be acknowledged. It was based on a single-center cohort and covered a lengthy period of time. Therefore, our scenario might not extend to the whole country, due to its heterogeneous epidemiology in terms of the etiologies of liver disease. Furthermore, even though we have carefully reviewed all the patients included in the database, some misclassifications of NASH as cryptogenic cirrhosis might have occurred, especially during the first years of observation.

In conclusion, our study showed that NASH significantly increased as an indication for WL registration in patients with cirrhosis, especially for those with HCC as the main indication for transplantation. The NASH recipients displayed good long-term post-operative survival. Although a NASH epidemic represents a serious health problem, institutions and countries are not well-prepared to address this in many fields of medicine [25], including also the liver transplant setting. The results provided by our study should be useful for improving our knowledge in this field and to prepare us to face this upcoming challenge.

## Figures and Tables

**Figure 1 medicina-58-00290-f001:**
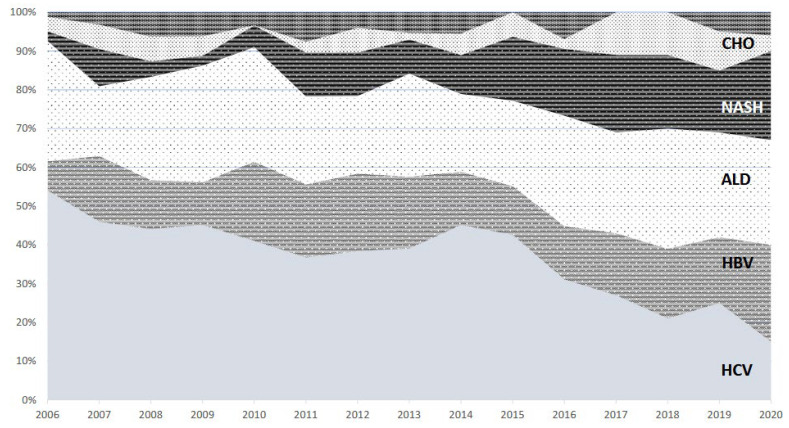
Temporal trends in waiting list registrations at Padua University Hospital Liver Transplant Center between 2006 and 06.2020. The field with a vertical b/w line accounts for other indications. Abbreviations. ALD: alcoholic liver disease; CHO: cholestatic/autoimmune disease; HBV: hepatitis B virus; HCV: hepatitis C virus, NASH: non-alcoholic steatohepatitis.

**Figure 2 medicina-58-00290-f002:**
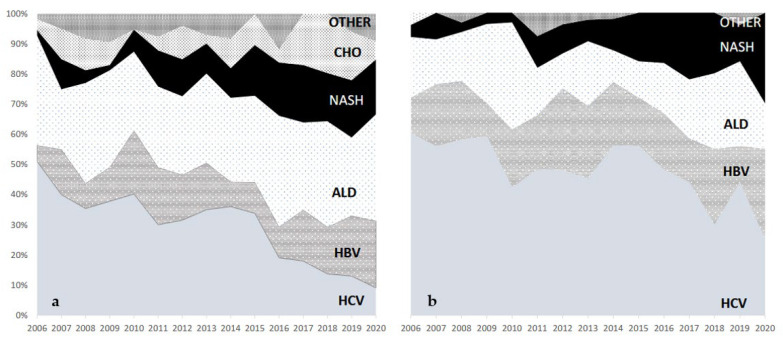
(**a**) Trends in waiting list registrations for patients with decompensated cirrhosis. (**b**) Trends in waiting list registrations for patients with hepatocellular carcinoma on compensated cirrhosis. The area depicted with a vertical b/w line accounts for other indications. Abbreviations. ALD: alcoholic liver disease; CHO: cholestatic/autoimmune disease; HBV: hepatitis B virus; HCV: hepatitis C virus; NASH: non-alcoholic steatohepatitis.

**Figure 3 medicina-58-00290-f003:**
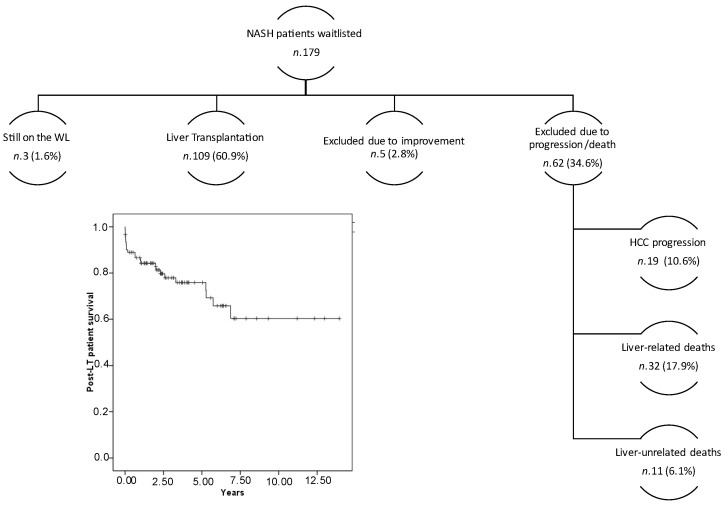
Outcomes of the waitlisted patients with NASH, and post-transplant survival in those who underwent liver transplantation.

**Table 1 medicina-58-00290-t001:** Clinical characteristics of the patients at the time of the waiting list, according to the underlying etiology. Abbreviations: BMI: body mass index; HCC: hepatocellular carcinoma; LT: liver transplantation; MELD: model for end stage liver disease; NASH: non-alcoholic steatohepatitis; WL: waiting list.

	Overall Cohort *n*. 1491	NASH *n*. 179	Non-NASH *n*. 1312	*p*-Value
Age, years	56 ± 9	59 ± 9	56 ± 9	<0.01
Gender, male, *n* (%)	1152 (77.2)	145 (81)	1007 (76.7)	0.2
BMI	25 ± 4	27 ± 5	25 ± 4	<0.01
HCC, *n*. (%)	754 (50.5)	96 (53.6)	658 (50.1)	0.4
Child-Pugh Score classes, *n*. (%) A B C	407 (27.3) 556 (37.3) 528 (35.4)	48 (26.8) 67 (37.4) 64 (35.7)	359 (27.3) 489 (37.2) 464 (35.4)	0.6
MELD score at WL registration	16 ± 7	16 ± 6	16 ± 7	1
Portal vein thrombosis	229 (15.3)	29 (16.2)	200 (15.2)	0.7
Refractory Ascites	365 (24.5)	48 (26.8)	317 (24.1)	0.4
Indication to LT - Decompensated cirrhosis ± HCC - MELD score at WL registration - HCC on compensated cirrhosis - MELD score at WL registration	920 (61.7) 19 ± 6 571 (38.3) 11 ± 3	111 (62) 19 ± 6 68 (38) 11 ± 3	809 (61.6) 19 ± 8 503 (38.3) 11 ± 4	0.9 0.8 0.9

## Data Availability

Not applicable.

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
