# Peer review of "Nash Up, Virus Down: How the Waiting List Is Changing for Liver Transplantation: A Single Center Experience from Italy"

_medicina, 2022, doi:10.3390/medicina58020290_

Round 1
Reviewer 1 Report
I read with great interest the manuscript entitled “NASH UP, VIRUS DOWN: HOW THE WAITING LIST IS CHANGING FOR LIVER TRANSPLANTATION: A SINGLE CENTER EXPERIENCE FROM ITALY.” However, I did not see any novelty in it. I think the authors should state the difference between Refs. 5, 16-18. Also, I think the authors should state the difference between Refs. 21 too.
Author Response
We want to the thank the Reviewer for his/her comment.
Our study aimed to retrospectively assess the increase of NASH as indication to liver transplantation in a single-center institution from Italy. In our paper, we demonstrated an increasing trend of NASH, as previously shown in other papers. Several differences should be considered, however. The studies by Younussi (Ref. 5), Goldberg (Ref. 16), Noureddin (Ref 18) and Wong (Ref 21) described the US scenario, whereas Calzadilla-Bertot et al. (Ref 17) described the trends in the frequency of NASH among adults listed and undergoing transplant in Australia and New Zealand.
The prevalence of NASH is heterogeneously reported worldwide. It seemed to be less pronounced in Southern Europe, where different prevalence in risk factor profiles, and presence of Mediterranean diet have been considered plausible protective factors in the past. Our study demonstrated for the first time that NASH related cirrhosis shows an increasing trend in the transplant setting also at our latitudes. As recently demonstrated, although NASH epidemic represents a serious health problem, Institutions and Countries are not well prepared to address it in many fields of Medicine (doi.org/10.1016/j.jhep.2021.10.025), including also the liver transplant setting. Indeed, as we mentioned in the Introduction section, patients with NASH related cirrhosis often share different (perhaps more serious) hepatic and extra-hepatic comorbidities than patients with other etiologies. Therefore, results provided by our study should be useful to improve our knowledge in this field and to get us prepared to face this upcoming challenge.
Reviewer 2 Report
The present study demonstrated the clinical characteristics of patients registered on the waiting list for liver transplantation. As indicating in the title, the patients with NASH increased rapidly while those with viral diseases decreased. The reviewer considers that the message of the manuscript is simple and clear and that the minor revision as described below is necessary.
- Is there any significance of the method of transplantation (Ex. living/deceased donor, blood type etc.) especially in the patients’ survival?
- The abbreviation should be displayed unabbreviated style when it appeared at the first time in the main text and figure legend.
- There are several typographical and grammatical errors in the present manuscript. The proofreading should be performed again.
Author Response
We want to thank the Reviewer for his/her insightful comments.
We considered only adult, deceased donor liver transplantations. None of the patients underwent AB0 incompatible liver transplant. We added this information in the revised version of the manuscript.
We apologize for the grammatical errors, the whole manuscript has been reviewed, as requested.
Reviewer 3 Report
The authors present an original work examining NASH UP, VIRUS DOWN: HOW THE WAITING LIST IS CHANGING FOR LIVER TRANSPLANTATION: A SINGLE CENTER EXPERIENCE FROM ITALY. It is well-written work. This manuscript has a potential to be published in this journal, however several points should be revised.
Major revisions
- At the part of the abstract, the last sentence is inadequate in the conclusion. Please clearly make sentences in conclusion.
- How many patients developed de novo NASH after liver transplantation? It is important to add the data of relationship between de novo NASH and cardiovascular events after liver transplantation. I agree that the trend of increasing NASH related liver disease undergoing liver transplantation is important issue.
- Please clarify the data of immunosuppressive regimen after liver transplantation for NASH. Prolonged steroid use might cause the metabolic syndrome after the surgery.
Minor revisions
- Table 1 is very confusing due to across the pages.
- The 3rd paragraph of the discussion, the beginning of the sentence is inadequate.
- Abbreviations of Figure 1&2 should be added full spellings.
Author Response
RESPONSE TO MAJOR REVISIONS
We want to thank the Reviewer for his/her insightful comments.
- We apologize for the mistake. The related sentence in the abstract section has been revised.
- We thank for the comment. The development of de novo NASH after transplantation was not an aim of our study, mainly for three reasons. First, we focused on patients with NASH related cirrhosis, whereas de novo NASH is by definition the development of post-LT NASH in patients with disease etiologies other than NASH in the pre-LT period. Second, diagnosis of post-LT NASH may require a prospective design, and often a histological confirmation. Third, this very important topic has been recently explored by our Group (1111/ctr.14532). In this prospective study, we have demonstrated that the prevalence of de novo metabolic syndrome was 46%, 43%, and 49% at 6, 12, and 24 months after LT, respectively.
- We thank for this very important comment. The most commonly used immunosuppressive regimens for patients with NASH were tacrolimus based, with MMF or everolimus as CNI-sparing agents. Steroids were not used as long-term immunosuppressive agents (both in patients with and without NASH as indication to LT). Induction with IL-2 RA was applied especially in critically ill patients. Finally, the immunosuppression schedule was tailored according to pre-LT and post-LT features (e.g., chronic kidney disease, post-LT sepsis, tacrolimus-induced side effects). A sentence has been added, accordingly.
RESPONSE TO MINOR REVISIONS
We thank for the comments. Table 1 is not across the pages in the revised version of the manuscript. Mistakes have been amended, accordingly. Abbreviations in Figs 1&2 have been added, as requested.
Round 2
Reviewer 1 Report
Thank you for your kind response. I understand what you want to say.
Since the prevalence of NASH varies by region, I think this point should be clearly mentioned in the introduction section.
I also think that this result should be mentioned in the discussion section as a limitation because it does not apply to all regions in this sense.
Author Response
We thank the Reviewer for his/her comments.
A specific sentence has been added both in the Introduction section and in the Discussion section, as requested.
We have already discussed the issue of heterogeneous presentation of NASH, which is a significant limitation of our study (please see page n. 7). Therefore, we agree with the Reviewer that data coming from the present paper could not be applied to all regions.
Best regards
Reviewer 3 Report
Thank you for giving me review this revised manuscript. Authors clearly answer the questions, and the content of the text is corrected according to the queries. This manuscript is accepted in its form. It is pleasure for me to have an opportunity to review this manuscript in this journal.
Author Response
We thank the Reviewer for his/her comments.
Best regards.